# Synthesis and Characterization of Bulky Substituted Hemihexaphyrazines Bearing 2,6-Diisopropylphenoxy Groups

**DOI:** 10.3390/molecules28155740

**Published:** 2023-07-29

**Authors:** Evgenii N. Ivanov, Verónica Almeida-Marrero, Oskar I. Koifman, Viktor V. Aleksandriiskii, Tomas Torres, Mikhail K. Islyaikin

**Affiliations:** 1IRLoN, Research Institute of Macroheterocycles, Ivanovo State University of Chemistry and Technology, 7, Sheremetievskiy Ave., 153000 Ivanovo, Russia; ivanov.isuct@gmail.com (E.N.I.); oik@isuct.ru (O.I.K.); nmr@isuct.ru (V.V.A.); 2G. A. Krestov Institute of Solution Chemistry of the Russian Academy of Sciences, 1 Akademicheskaya Str., 153045 Ivanovo, Russia; 3Department of Organic Chemistry, Autonoma University of Madrid, Cantoblanco, 28049 Madrid, Spain; veronicaalmeidamarrero@gmail.com; 4Institute for Advanced Research in Chemical Sciences (IAdChem), Autonoma University of Madrid, 28049 Madrid, Spain; 5Instituto Madrileño de Estudios Avanzados (IMDEA)—Nanociencia, c/Faraday 9, Cantoblanco, 28049 Madrid, Spain

**Keywords:** 2,5-diamino-1,3,4-thiadiazole, hemihexaphyrazines, expanded hemiporphyrazines

## Abstract

New substituted [30]trithiadodecaazahexaphyrines (hemihexaphyrazines) were synthesized by a crossover condensation of 2,5-diamino-1,3,4-thiadiazole with 4-chloro-5-(2,6-diisopropylphenoxy)- or 4,5-bis-(2,6-diisopropylphenoxy)phthalonitriles. The compounds were characterized by ^1^H-, ^13^C-NMR, including COSY, HMBC, and HSQC spectroscopy, MALDI TOF spectrometry, elemental analysis, IR and UV-Vis absorbance and fluorescence techniques.

## 1. Introduction

Porphyrinoids have emerged as attractive molecular building blocks for arrangement into molecular materials and nanotechnological devices [1,2,3,4,5]. To date, they have been successfully incorporated by us as active components in photo- and electroactive systems for optoelectronics [6,7,8,9], solar energy conversion [10,11,12,13], and biomedicine [14,15,16,17], among others.

Hemihexaphyrazines (Hhps), are a class of macrocyclic compounds that exhibit unique chemical properties and have gained significant attention in various scientific fields. These compounds consist of nitrogen, sulfur, and carbon atoms arranged in a highly symmetrical and complex structure, including three thiadiazole and three isoindole units linked together in an alternating fashion via nitrogen atoms to form a six-member macroheterocyclic system (see **4**, for example). They can be classified as expanded hemiporphyrazines [18]. Their large conjugated systems enable efficient absorption and emission of light. Moreover, Hhps exhibit tunable redox behavior, which can be exploited in energy storage systems and electrochemical devices. Their ability to coordinate with metal ions allows for the creation of functional metallo-Hhp complexes with potential applications in molecular recognition and sensing.

Their structure was unequivocally established by us by gas-phase electron diffraction [19,20,21] and X-ray data [22]. It was revealed that H_3_Hhp is characterized by extremely high thermal stability [23] and is able to form long self-organized rows on the surface of Au(111), which assumes a space-controlling deprotonating process, and thereby shows potential as a new material for information storage [24]. Hhps have an expanded inner cavity and therefore are able to accommodate three metallic atoms [25,26,27]. Recently it was shown that due to the presence of nine nitrogen atoms in the inner ring, H_3_Hhp can form a double-decker complex with potassium [28]. Metallation of the macrocycle with diethylamid lithium led to another unusual double-decker structure in which two Li_3_Hhp are joined by Cl atom [29].

Hhps are flat expanded macrocycles with high structural versatility, including multiple modifications by the introduction of peripheral groups and the incorporation of various metal atoms in the central cavity. It was established that homotrinuclear Ni- and Cu-complexes of Hhp can be reduced in anaerobic conditions to produce dianion radicals with interesting magnetic properties [30,31]. These compounds show promising properties of great interest as components of molecular conjugates with other photo- and electroactive species. However, as free, unsubstituted bases, H_3_Hhps are very poorly soluble in organic solvents, which results in aggregation phenomena that can be obviated by the introduction of bulky peripheral substituents, facilitating their synthetic use in functionalization processes [32,33]. Recently we have reported on the synthesis of hexa(3,6-hexyl)hemihexaphyrazine [34].

Among these bulky substituents, diisopropylphenoxy groups have been frequently used in Hph-related porphyrinoides because they provide high macrocycle solubility and dramatically reduce macrocycle aggregation [35,36,37,38,39,40]. For this reason, in this work, we propose the preparation of new Hphs with bulky substituents from precursors such as 4,5-bis(2′,6′-diisopropylphenoxy)- and 4-(2′,6′-diisopropylphenoxy)-5-chlorophthalonitriles. The latter would allow the post-functionalization of the macrocycle by reactions on the chlorine atoms.

## 2. Results and Discussion

Substituted phthalonitriles **1** and **2** were synthesized from commercially available 2,6-diisopropylphenol and 4,5-dichlorophthalonitrile according to reported procedures [37,38]. The compounds were characterized by ^1^H-NMR, and correct assignment of signals of both was necessary for achieving the proper assignation of the protons of the target macrocycles **3** and **4**. 2,5-Diamino-1,3,4-thiadiazole was prepared according to a known procedure [41]. Substituted H_3_Hhps **3** and **4** were prepared by a crossover condensation of the corresponding phthalonitriles with an equimolar amount of 2,5-diamino-1,3,4-thiadiazole in anhydrous ethylene glycol in an argon atmosphere at reflux temperature (Figure 1). Compound **3** consists of a mixture of two regioisomers with symmetries C_1_ and C_3_, respectively, due to the asymmetry of the starting phthalonitrile **1,** which could not be separated.

Column chromatography in silica gel using a mixture of heptane/ethyl acetate (5:1) as an eluent was applied to yield intensely orange-colored macroheterocycles **3** and **4**. It was found that due to the presence of the bulky substituents on the periphery, **3** and **4** were highly soluble in common organic solvents such as DCM, CHCl_3_, THF, ethyl acetate, acetone and toluene at room temperature. Compounds **3** and **4** were characterized by MS (MALDI-TOF), UV-Vis, IR, NMR and elemental analysis.

The MS spectrum of **3** (Figure 1) shows a molecular ion located at 1314.6 *m*/*z*, which corresponds to protonated form [**3** + H]^+^, along with signals of lower intensities at 1336.4 *m*/*z* and 1352.6 *m*/*z*, corresponding to [**3** + Na]^+^ and [**3** + K]^+^ ions, respectively.

A peak at 1738.8 *m*/*z* (Figure 2) that corresponds to molecular ion [**4** + H]^+^ was detected in the mass spectrum of **4** along with signals of lower intensity at 1760.7 and 1776.7 *m*/*z* corresponding to [**4** + Na]^+^ and [**4** + K]^+^ ions.

Due to their high solubility in organic solvents, these compounds are of great interest as subjects for NMR spectroscopy studies. ^1^H-NMR spectra of **3** and **4** recorded in CDCl_3_ are showed in Figure 3 and Figure 4, respectively. One can distinguish three principal areas of the signal location: 0.5–3.2 ppm—protons of aliphatic isopropyl groups; 6.5–8.1 ppm—protons of aromatic systems; and 12.0–12.5 ppm—portions of intrinsic N-H groups. Positions and integrals of the proton signals of the first two groups of signals are in good agreement with data described previously for the aliphatic and aromatic parts of porphyrinoids bearing bulky groups [32,33,42]. Hence, in comparison with octasubstituted phthalocyanine [37], where the signal of protons of inner imino groups was found to be located in a high field (−0.53 ppm), the corresponding signals of H_3_Hhps **3** and **4** were found in a low field, ca. 12 ppm. The appearance of these signals in a low field is typical of hemiporphyrazine free bases and confirms the nonaromatic character of the ABABAB macrocyclic system.

It is worth noting that a singlet at 12.24 ppm in the ^1^H-NMR spectrum of **4** (Figure 4) is split into two signals (12.35 and 12.40 ppm) for **3** due to its lower symmetry. The presence of these two signals in the spectrum of **3** can be explained by the formation of the C_1_ and C_3_ regioisomers. Previously, the same effect was observed for related camphor-substituted H_3_Hhps [32]. The integrals of these two signals can be used to estimate the ratio of the C_1_ and C_3_ regioisomers at 3:2.

For a comprehensive structural characterization of compound **4**, various NMR spectra including ^1^H-, ^13^C-NMR, 2D-correlations COSY ^1^H-^1^H, HSQC ^1^H-^13^C and HMBC ^1^H-^13^C were performed (Appendix A). To accurately assign the signals in the ^1^H, ^13^C NMR spectra, quantum chemical calculations of the magnetic shielding constants were performed using the GIAO method based on the optimized structure of **4** obtained through DFT CAM-B3LYP 6-31G(d,p) calculations. Notably, a high level of agreement between the experimental and calculated chemical shifts was observed (Appendix A).

The analysis of the COSY spectrum (Appendix A) revealed the presence of cross-peaks originating from the interaction between protons CH (3.08 ppm) and CH_3_ (1.21, 1.15 ppm) of the diisopropyl fragments. Additionally, the cross-peaks of the phenyl protons overlapped with the diagonal signals. In the HMBC ^1^H-^13^C correlation spectrum (Appendix A), cross-peaks between protons b, c, d, e, f (as denoted in Figure 4) and the corresponding carbon atoms were observed. Moreover, in the HMBC ^1^H-^13^C spectrum (Appendix A), cross-peaks resulting from the interaction between NH protons (12.24 ppm) and carbon atoms (128.5, 152.82 ppm) of the pyrrole fragment were noteworthy.

The aromaticity of the macrorings, which form the foundation of porphyrinoids, is a crucial aspect in the chemistry of these compound families. The optimized geometry of compound **4** reveals a planar framework consisting of three thiadiazole and isoindole rings connected by nitrogen atoms. The phenyl rings of the lateral substituents, specifically the 2,6-diisopropylphenoxyl groups, exhibit a rotational orientation with respect to the macrocyclic plane of approximately 76 degrees (+0.22/−0.07). Overall, the structure of molecule **4** exhibits approximate D_3_ point group symmetry.

The evaluation of global and local aromaticity was conducted using the GIAO/CAM-B3LYP/6-31G(d,p) method based on the geometry optimized at the DFT/CAM-B3LYP/6-31G(d,p) level. The results of these calculations are presented in Table 1. To assess the impact of bulky substituents on aromaticity, the nonsubstituted H_3_Hhp molecule was also calculated using the same methodology (Table 1).

A positive NICS (nucleus-independent chemical shift) value of 1.52 ppm was observed at the mass center A of molecule **4**, indicating its nonaromatic nature. Additionally, the calculated chemical shifts near the exocyclic N-atoms B exhibited low negative values, suggesting weak electron circulation, and further supporting the nonaromatic character of the macrocycle. Furthermore, a significant alternation (0.085 Å) was observed in the bond lengths of the exocyclic atoms N (N_ex_-C_thia_ (1.369 Å) and N_ex_-C_pyrr_ (1.284 Å), consistent with previous findings [22] for hexapentoxyhemihexaphyrazine. These results provide additional evidence for the nonaromatic nature of the macrocycle based on hemihexaphyrazine.

It is worth noting that the pyrrole moieties (centers C) within the isoindoline subunits lose their aromaticity due to the presence of double bonds connecting them to the exocyclic nitrogen atoms. This structural modification significantly disrupts their local aromaticity. However, the aromaticity of the benzene rings (centers D), thiadiazole rings (centers E), and benzene cycles (centers F) in the lateral substituents remains preserved.

As shown in Table 1, the introduction of bulky substituents at the periphery of the macrocycle has minimal impact on the aromaticity of compound **4**.

The UV-Vis and emission spectrum of **3** and **4** were recorded in CHCl_3_ (Figure 5). The shape of the spectral curve of the spectrum is typical for the ABABAB family of macrocycles [24,25,41]. The location of the absorption maxima in the violet part of visible spectrum confirms the nonaromatic character of the compounds.

Measurements by emission spectroscopy of compounds **3** and **4** in chloroform were carried out at room temperature using excitation by visible light, the wavelengths of which correspond to absorption maxima of 419 and 423 nm, respectively. It was established that **4** generates fairly broad fluorescence spectra that maximize around 600 nm (Figure 5), giving rise to a virtual mirror image with different intensities than in the absorption spectrum. The fluorescence quantum yields of **3** and **4** in CHCl_3_ were found to be equal to 0.050 and 0.084 respectively. It is worth noting that the characteristics found are in agreement with those revealed earlier for pentoxy-substituted H_3_Hhps [22], and their low values indicate the essential participation of radiationless channels for quenching of excited states. The huge values of the Stoke’s shift (161 nm for **3**, 158 nm for **4**) show that essential structural rearrangements take place when the molecules are in the excited states. The reasons for this are under study.

Views of molecular orbitals of **4** are shown in Figure 6.

## 3. Experimental Section

### 3.1. Materials and Methods

Inert conditions and standard glassware were used to perform all reactions, which were monitored using TLC plates pre-coated with silica gel 60-F254 (Merck). Column chromatography was carried out using Merck silica gel 40–63 μm, 230–400 mesh and Fluka silica gel, 40–200 mesh. ^1^H-NMR spectra were performed using Bruker DRX 500, Bruker Avance and Bruker Avance II (300 and 500 MHz) spectrometers furnished by the Interdepartmental Investigation Service (SIdI) of the Universidad Autónoma de Madrid (UAM). Internal references for all spectra were established using the residual solvent of CDCl_3_ (^1^H: δ = 7.26), relative to SiMe_4_. ^13^C-NMR, 2D spectra were performed using an Avance III Bruker 500 NMR spectrometer furnished by the Joint Research Center, Upper Volga Regional Center of Physical and Chemical Research, Ivanovo, at operating frequencies of 500.17, 125.77 MHz, respectively. A 5 mm 1H/31P/D-BBz-GRD Triple Resonance Broad Band Probe (TBI) was employed. The standard pulse sequence WALTZ 16 from the TopSpin 3.6.1 software was used for ^13^C{^1^H} NMR spectra registration. There were 16,000 scans in the spectral range of 29761.9 Hz with a power of RG amplifier (RG = 2050); 32,768 data points were acquired. To assign the NMR signals in the ^1^H- and ^13^C-spectra, the two-dimensional methods COSY, HSQC and HMBC were used. Temperature control was achieved using a Bruker variable temperature unit (BVT-2000) in combination with a Bruker cooling unit (BCU-05) to provide chilled air. Experiments were run at 298 K without sample spinning. The inaccuracy of the chemical shift measurement with respect to the external standard, HMDSO (Sigma Aldrich, St. Louis, MO, USA), was evaluated as ±0.01 ppm for ^1^H and ±0.1 ppm for ^13^C NMR spectra. 

The two-dimensional correlation spectroscopy (2D COSY) spectra with a zero-quantum suppression element were acquired with a 16.96 ppm spectral window in the direct dimension F1 with 2048 complex data points and a 16.96 ppm spectral window in the indirect dimension F2 with 128 complex points. The spectra were acquired with 64 scans and relaxation delay of 2 s.

The 2D ^1^H-^13^C HSQC (^1^H-^13^C correlation via double INEPT transfer) spectra were recorded in a phase-sensitive mode using the Echo/Antiecho-TPPI gradient selection with decoupling during acquisition.

The 2D ^1^H-^13^C HMBC correlation via heteronuclear zero and double quantum coherence optimized on long-range couplings (no decoupling during acquisition) using gradient pulses for selection were recorded using «Hmbcgpndqf» (TopSpin3.6.1). JASCO V-660 and JASCO FP-8600 spectrophotometers were used to measure UV-Vis and fluorescence, respectively, in the Department of Organic Chemistry at Universidad Autónoma de Madrid. Matrix-assisted laser desorption/ionization time of flight (MALDI-TOF) was recorded using a AXIMA Confidence Shimadzu spectrometer, elemental analysis was performed on a Flach EA 1112 instrument, and IR spectra were carried out on an Avatar 360 FT-IR ESP spectrophotometer using the resources of the Center for Collective Use of Scientific Equipment of Ivanovo State University of Chemistry and Technology. Fluorescence quantum yields were determined as reported in the literature [43], using tetraphenylporphyn (TPP) as a standard.

### 3.2. Synthesis

4-Chloro-5-(2,6-diisopropylphenoxy) phthalonitrile (**1**) [37]. Anhydrous potassium carbonate (2.4 g, 15.5 mmol) was added to a solution of 2,6-diisopropylphenol (0.9 g, 5 mmol) and 4,5-dichlorophthalonitrile (0.98 g, 5 mmol) in dry DMF (75 mL). The reaction mixture was heated in an argon atmosphere at 45 °C for 24 h. It was then cooled down and poured into water and the precipitate was filtered off and washed with water. After drying, the crude product was purified by column chromatography using a mixture of heptane/ethyl acetate (5:1) as an eluent. Yield: 64% (1.09 g); mp 181 °C. ^1^H-NMR (300 MHz, CDCl_3_): δ (ppm) = 7.83 (s, H), 7.32–7.21 (m, 3 H), 6.68 (s, H), 2.68 (sept, *J* = 6.9 Hz, 2H), 1.12 (d, *J* = 6.9 Hz, 12H).

4,5-Bis(2,6-diisopropylphenoxy) phthalonitrile (**2**) [38]. Anhydrous potassium carbonate (2.4 g, 15.5 mmol) was added to a solution of 2,6-diisopropylphenol (1.8 g, 10 mmol) and 4,5-dichlorophthalonitrile (0.5 g, 2.5 mmol) in dry DMF (40 mL). The reaction mixture was heated at 80 °C in an argon atmosphere for 48 h. It was then cooled down and poured into water and the precipitate was filtered off and washed with water. After drying, it was purified by column chromatography using a mixture of heptane/ethyl acetate (5:1) as an eluent. Yield: 34% (0.41 g); mp 179-180 °C. ^1^H-NMR (300 MHz, CDCl_3_): δ (ppm) = 7.31 − 7.19 (m, 6H), 6.68 (s, 2H), 2.88 (sept, *J* = 6.9 Hz, 4H), 1.15 (d, *J* = 6.9 Hz, 24H).

2,14,26-Trichloro-3,15,27-tri[2′,6′-diisopropylphenoxy]-5,36:12,17: 24,29-triimino-7,10: 19,22: 31,34–tritio-[*f*,*p*,*z*]–tribenzo-1,2,4,9,11,12,14,19,21,22,24,29-dodecazacyclotriaconta-2,4,6,8,10,12,14,16,18,20,22,24,26,28,30-pentadecaene (**3**)

A mixture of 4-chloro-5-(2,6-diisopropylphenoxy)phthalonitrile (0.58 g 0.72 mmol) and 2,5-diamino-1,3,4-thiadiazole (0.2 g 1,72 mmol) in 10 mL of anhydrous ethylene glycol was heated at reflux during 24 h in an argon atmosphere. Water was added to the solution, resulting in the formation of a precipitate that was filtered off, washed with water, and extracted with CHCl_3_ after drying. The solution was dried over MgSO_4_ and filtered off, and the solvent was evaporated under reduced pressure. Final purification was performed by column chromatography using silica gel as a solid phase and a mixture of heptane/ethyl acetate (5:1) as an eluent, resulting in an orange crystalline solid. Yield: 24% (0.18 g); mp > 240 °C. ^1^H-NMR (500 MHz, CDCl_3_): δ (ppm) = 12.40–12.35 (m, 3H) 8.06–7.98 (m, 3H), 7.27–7.19 (m, 8H), 7.12–7.11 (m, 3H), 6.97–6.94 (m, 3H), 2.93–2.88 (m, 6H), 1.19–1.01 (m, 36H). UV-Vis (CHCl_3_) λ_max_ nm (log *ε*, dm^3^∙mol^−1^∙cm^−1^): 397 (4.89), 419 (4.91), 467 (4.12), 507 (3.76). IR (KBr) ν (cm^−1^): 3434, 3225, 2936, 2926, 2860, 1627, 1433, 1364, 1257, 1209, 1065, 965, 534. MS (MALDI-TOF, CHCA), *m*/*z*: 1314.6 [M + H]^+^, 1336.4 [M + Na]^+^, 1352.6 [M + K]^+^.

2,3,14,15,26,27-Hexa[2′,6′-diisopropylphenoxy]-5,36:12,17:24,29-triimino-7,10:19,22:31,34-trithio-[*f*,*p*,*z*]-tribenzo-1,2,4,9,11,12,14,19,21,22,24,29-dodecazacyclotriaconta-2,4,6,8,10,12,14,16,18,20,22,24,26,28,30-pentadecaene (**4**) 

A mixture of 4,5-bis-(2,6-diisopropylphenoxy) phthalonitrile (0.4 g 0.833 mmol) and 2,5-diamino-1,3,4-thiadiazole (0.096 g 0.833 mmol) in 10 mL of anhydrous ethylene glycol was heated at reflux during 24 h in an argon atmosphere. The reaction mixture was added to water, resulting in the formation of a precipitate that was filtered off, washed with water, and extracted with CHCl_3_ after drying. The solution was dried over MgSO_4_ and filtered off, and the solvent was evaporated under reduced pressure. Final purification was performed by column chromatography using silica gel as a solid phase and a mixture of heptane/ethyl acetate (5:1) as an eluent, resulting in an orange crystalline solid. Yield: 17% (0.08 g) mp > 240 °C. ^1^H-NMR (500 MHz, CDCl_3_): δ (ppm) = 12.24 (s, 3H), 7.33–7.19 (m, 24H), 3.13–3.04 (m, 12H), 1.22–1.12 (m, 72H). ^13^C-NMR (125 MHz, CDCl_3_): δ (ppm) = 169.77, 152.82, 152.17, 148.27, 141.38, 128.49, 126.63, 124.88, 107.9, 27.5, 24.24,22.73.

UV-Vis (CHCl_3_) λ_max_ nm (log *ε*, dm^3^∙mol^−1^∙cm^−1^): 292 (4.80) 400 (4.96), 423 (4.91), 466 (4.37), 509 (4.16). IR (KBr) ν (cm^−1^): 3410, 2963, 2924, 2861, 1620, 1480, 1444, 1364, 1376, 1274, 991, 887, 850, 480. MS (MALDI-TOF, CHCA), *m*/*z*: 1738.8 [M + H]^+^, 1760.7 [M + Na]^+^, 1776.7 [M + K]^+^. 

### 3.3. Quantum Chemical Calculations

Geometry optimization of **4** was carried out using density functional theory (DFT) calculations utilizing long-range corrected hybrid functional CAM-B3LYP [44] with a 6-31G(d,p) basis set. Force field calculations performed at the same level indicated no imaginary frequencies. All calculations were performed using Gaussian 16 software [45]. Optimized geometry parameters are shown in Appendix A.

To account for solvation effects, the NMR shielding constants were calculated using the GIAO method [46] using the polarizable continuum model (PCM). To ensure accurate calculations, benzene and TMS (tetramethylsilane) were selected as the standards for determining the chemical shifts of sp^2^- and sp^3^-hybridized carbons, respectively, following established recommendations [47,48]. The geometry and shielding parameters of the reference compounds were calculated using the same theoretical approach as the compounds under investigation.
$$\delta_{i}=\sigma_{ref}-\sigma_{i}+\delta_{ref}$$
where *σ_ref_*, *σ_i_* represent the shielding constants calculated for **4** and standards, and *δ_ref_* is an experimental chemical shift of the reference compound (128.5 ppm for benzene ^13^C NMR, 0 ppm for TMS). Correlations between experimental (d_exp_) and computational (d_calc_) (GIAO) ^13^C and ^1^H chemical shifts (**4**) are shown in Appendix A. Calculations of the nucleus-independent chemical shift (NICS) [49] were performed for structure **4**.

## 4. Conclusions

Bulky substituted trichlorotri(2,6-diisopropylphenoxy)- and hexa(2,6-diisopropylphenoxy) hemihexaphyrazines **3** and **4** were prepared for the first time by condensation of 4-chloro-5-(2,6–diisopropylphenoxy) phthalonitrile and 4,5-bis-(2,6–diisopropylphenoxy) phthalonitrile, respectively, with 2,5-diamino-1,3,4-thiadiazole using ethylene glycol as solvent. Their high solubility and lack of aggregation in organic solvents allowed easy purification by column chromatography and spectroscopic characterization, which is unusual with these kinds of porphyrinoids. The compounds were characterized by IR, NMR, absorption and emission UV-Vis spectroscopy, and mass spectrometry. An expanded inner cavity endowed with 15 nitrogen and 12 carbon atoms provides these systems with unique coordination properties, which will be reported in due course. 

## Data Availability

Not applicable.

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
