# Peer review of "Synthesis and Characterization of Bulky Substituted Hemihexaphyrazines Bearing 2,6-Diisopropylphenoxy Groups"

_molecules, 2023, doi:10.3390/molecules28155740_

Round 1
Reviewer 1 Report
New compounds, substituted [30]trithiadodecaazahexaphyrines (hemihexaphyrazines), were successfully synthesized. The compounds were characterized by 1H-, 13C-NMR, including COSY, HMBC, and HSQC spectroscopy, MALDI TOF spectrometry, elemental analysis, IR and UV-Vis absorbance and fluorescence techniques.
I recommend publication after attention to the matters outlined below.
Compound 1
I think the signal marked "m" is sept.
2.73-2.63 (m, J = 6.9 Hz, 2H) -> 2.68 (sept, J = 6.9 Hz, 2H)?
The signal marked "dd" is d or two d signals, I think.
1.16-1.05 (dd, J = 6.9 Hz, 12H) -> 1.15 (d, J = 6.9 Hz, 12H) or X.XX (d, J = 6.9 Hz, 6H), Y.YY (d, J = 6.9 Hz, 6H)
Compound 2
2.94-2.81 (m, J = 6.9 Hz, 4H) -> X.XX (sept, J = 6.9 Hz, 4H)
1.21-1.09 (dd, J = 6.9 Hz, 24H) -> X.XX (d, J = 6.9 Hz, 24H) or Y.YY (d, J = 6.9 Hz, 12H), Z.ZZ (d, J = 6.9 Hz, 12H).
If 1H NMR spectra were complicated, these spectroscopic results might be the restricted rotations of the bonds between the carbon atoms of Dip groups and the O atoms of the phthalonitrile skeleton in 1 or 2, caused by the steric congestion of 2,6-diisopropylphenyl groups. To solve this problem, measurement of various temperatures 1H NMR might be useful.
It is preferable to write NICS(0) not NICS.
For fluorescence spectra, the wavelength of the excitation light should be noted.
Author Response
"Please see the attachment."

Reviewer 2 Report
This is a nicely written manuscript describing an interesting and novel macrocycle synthesis. I am happy to recommend acceptance. The authors may consider to include the following relevant references:
The original reference for the phthalonitrile and its use in phthalocyanine formation should be included: Angewandte Chemie, International Edition (2005), 44(46), 7546-7549
A sub-phthalocyanine derived from the same phthalonitrile, of similar symmetry as the current macrocycle, has been reported: Journal of Porphyrins and Phthalocyanines (2016), 20(8/11), 1034-1040
Author Response
"Please see the attachment."

Reviewer 3 Report
1- The melting points of compounds should be given in main text.
2- FT-IR spectra of the compounds should be given in Suppl. file and discussed/referred to such as Haloalkanes and aromatic hydrocarbons sensing using Langmuir–Blodgett thin film of pillar [5] arene-biphenylcarboxylic acid. Synthesis and photophysical properties of modifiable single, dual, and triple-boron dipyrromethene (Bodipy) complexes.
3- The figures are in various forms. Please, Adjust uni-form. is not big-small-flu.
4- Give H-NMR values with splitting amount and CH2 or CH3 or etc.
Best regards
Author Response
"Please see the attachment."
